# Antioxidant Versus Pro-Apoptotic Effects of Mushroom-Enriched Diets on Mitochondria in Liver Disease

**DOI:** 10.3390/ijms20163987

**Published:** 2019-08-16

**Authors:** Adriana Fontes, Mireia Alemany-Pagès, Paulo J. Oliveira, João Ramalho-Santos, Hans Zischka, Anabela Marisa Azul

**Affiliations:** 1Institute of Molecular Toxicology and Pharmacology, Helmholtz Center Munich, German Research Center for Environmental Health, D-85764 Neuherberg, Germany; 2CNC-Center for Neuroscience and Cell Biology, University of Coimbra, 3004-504 Coimbra, Portugal; 3DCV-Department of Life Sciences, Faculty of Sciences and Technology of the University of Coimbra, 3000-456 Coimbra, Portugal; 4IIIUC-Institute for Interdisciplinary Research, University of Coimbra, 3030-789 Coimbra, Portugal; 5Institute of Toxicology and Environmental Hygiene, Technical University Munich, D-80802 Munich, Germany

**Keywords:** mitochondria, non-alcoholic fatty liver disease, fungi, mushrooms, truffles, antioxidants, oxidative stress, lipid metabolism, apoptosis, NASH, HCC

## Abstract

Mitochondria play a central role in non-alcoholic fatty liver disease (NAFLD) progression and in the control of cell death signalling during the progression to hepatocellular carcinoma (HCC). Associated with the metabolic syndrome, NAFLD is mostly driven by insulin-resistant white adipose tissue lipolysis that results in an increased hepatic fatty acid influx and the ectopic accumulation of fat in the liver. Upregulation of beta-oxidation as one compensatory mechanism leads to an increase in mitochondrial tricarboxylic acid cycle flux and ATP generation. The progression of NAFLD is associated with alterations in the mitochondrial molecular composition and respiratory capacity, which increases their vulnerability to different stressors, including calcium and pro-inflammatory molecules, which result in an increased generation of reactive oxygen species (ROS) that, altogether, may ultimately lead to mitochondrial dysfunction. This may activate further pro-inflammatory pathways involved in the progression from steatosis to steatohepatitis (NASH). Mushroom-enriched diets, or the administration of their isolated bioactive compounds, have been shown to display beneficial effects on insulin resistance, hepatic steatosis, oxidative stress, and inflammation by regulating nutrient uptake and lipid metabolism as well as modulating the antioxidant activity of the cell. In addition, the gut microbiota has also been described to be modulated by mushroom bioactive molecules, with implications in reducing liver inflammation during NAFLD progression. Dietary mushroom extracts have been reported to have anti-tumorigenic properties and to induce cell-death via the mitochondrial apoptosis pathway. This calls for particular attention to the potential therapeutic properties of these natural compounds which may push the development of novel pharmacological options to treat NASH and HCC. We here review the diverse effects of mushroom-enriched diets in liver disease, emphasizing those effects that are dependent on mitochondria.

## 1. Introduction

Non-Alcoholic Fatty Liver Disease (NAFLD) is a metabolic condition characterized by the accumulation of fat in more than 5% of the liver parenchyma, which is observed in the absence of other recognized causes of fatty liver (alcohol, chronic viral infection, drugs, autoimmunity, etc.) [1]. NAFLD can progress from isolated macro- or microvesicular steatosis (NAFL) to non-alcoholic steatohepatitis (NASH), a pathological state characterized by steatosis, inflammation and hepatocellular ballooning [2]. Whilst adverse NAFLD is estimated to affect 25% of the global population, NASH, with an estimated worldwide prevalence of 1–3%, is considered to be the initial pathological condition in progressive liver disease. The development of fibrosis following the activation of wound healing/repair mechanisms in response to chronic hepatic injury is a common feature at this stage, affecting 40–50% of NASH patients. In a 7–8-year period, 4–25% of individuals with NASH progress to cirrhosis as a consequence of the histological remodelling upon chronic fibrogenic processes [3]. Concomitantly, NASH increases the risk for hepatocellular carcinoma (HCC), even independently of cirrhosis, and whilst cardiovascular diseases (CVDs) represent the main cause of death in NAFLD patients, an increase in liver-related mortality is expected in upcoming years. NASH is currently the third most common indication for liver transplantation in the United States and accounts for 10% of all HCC cases in Europe [4,5,6,7].

Regarded as the hepatic manifestation of the Metabolic Syndrome (MetS), NAFLD is driven by excessive energy intake and/or reduced energy expenditure. In an attempt to normalize its regular metabolism and respond to the ectopic accumulation of fat, the liver triggers a process of bioenergetic remodelling that over time can become pathological, leading to the progression of NASH [1,7,8]. Hepatic mitochondria, which are at the centre of hepatocellular metabolism, are known to be adversely affected in NAFLD [9]. The increase in nutrient availability causes systemic metabolic alterations that lead to an increase in hepatic mitochondrial respiration as well as changes in the mitochondrial lipid membrane composition. These structural and functional alterations might increase the vulnerability to additional stressors. An increased generation of Reactive Oxygen Species (ROS) accompanied by lower antioxidant capacity is observed in late-stages of mitochondrial dysfunction, that in turn might contribute to inflammation and cell death [9,10,11,12]. Thus, ameliorating mitochondrial dysfunction may represent an efficient intervention to delay the progression of NASH [13,14].

Mushrooms and truffles (Box 1) have been part of culinary culture since antiquity (at least 40,000 years ago) and used in traditional Chinese medicine for dozens of centuries [15,16]. Later adopted by western medicine, mushrooms are now emerging as possible natural products for the treatment of conditions such as obesity, Type 2 Diabetes Mellitus (T2DM), CVDs and also NAFLD [16,17,18,19]. Mushrooms are a rich source of bioactive compounds known for their modulatory activities on the gut microbiota and enteric absorption, their antioxidant activity and their pro-apoptotic action [20,21,22,23,24,25]. These pleiotropic effects on metabolism and their influence on mitochondrial function might call upon mushroom compounds as interesting candidates for NASH and HCC therapeutics.

Box 1The fungi kingdom (yeasts, molds and mushrooms) encompasses a large and diverse group of living organisms.Macrofungi are higher fungi, mostly belonging to the divisions Ascomycota and Basidiomycota, with fruiting bodies producing spores in a distinct structure called “ascoma” or “basidiome”, respectively, which can grow above ground (mushrooms) and below ground (truffles). Fruiting bodies have multicellular structures with differentiated tissues developed from a mycelium, the vegetative part of a fungus, that consists of a network of interconnected hyphae. Macrofungi have different lifestyles, some establish a symbiotic association with roots of plants (called mycorrhiza), others are saprophytic and decompose dead organic material; very few are parasitic [26]. Mushrooms and truffles are an excellent source of polysaccharides (α/ß-D-glucans) [27], proteins [28,29], vitamins (B1, B2, B12, C, D, and E), minerals, and essential amino acids; they are also low in total fat but have a high content of polyunsaturated fatty acids. They are also rich in phenolic acids, such as hydroxybenzoic and hydroxycinnamic acid, flavonoids, tocopherols, ascorbic acid and carotenoids that are known for their antioxidant activity [24,30,31]. Some mushrooms have been traditionally used for centuries in diet or as medicine, such as the medicinal species *Ganoderma lucidum *(Curtis) P. Karst (reishi/lingzhi), which has been extensively studied for its biological activities, which have been associated with more than 200 metabolites identified so far [32]. Bioactive molecules from mushrooms (and truffles) have anticholesterolemic, anti-obesogenic, antihyperglycemic, cardiovascular protector, hepatoprotective, immunomodulatory, antitumour, antiviral, antibacterial, antiparasitic, antifungal, antiallergic, and detoxification effects [17,18,19]. About 1154 edible and food mushrooms and truffles were reported in 85 countries, in a total of 2327 wild useful species [33]. However, only very few species (ca. 25) are valued commercially, the cultivated *Agaricus *spp. (champignons); *Ganoderma *spp.; *Lentinus edodes *(Berk.) Pegler (shiitake); *Pleurotus *spp. (oyster mushroom); and the edible mycorrhizal species that include the *Boletus *spp. (porcini mushrooms), *Cantharellus cibarius *Fr. (golden chanterelle mushroom), *Lactarius deliciosus* (L.:Fr.) S.F. Gray (saffron milk cap), *Tricholoma matsutake *(S. Ito et S. Imai) Singer (matsutake), *Terfezia *spp. (desert truffles) or *Tuber *spp. (truffles) [34,35]. The scientific name of fungi follows the International Code of Nomenclature for Fungi and is composed by a Latinised binomial in italics representing the genus and the species name, followed by the name of the author(s) that first described them and coined the name. As such, the truncated form of the surname of famous fungi pioneers follows the fungi epithet (e.g. Fries (Fr.), Persoon (Pers.), Singer (Sing.), Paul Kummer (P. Kumm), Quélet (Quél.), Bulliard (Bull.)) and in instances of taxonomic modifications, namely relocation of species to another genus, the authority responsible for the first taxonomic grouping in brackets is followed by the authority performing the genus amendment [36]. Intimately associated to cultural heritage, fungi are also known for their common name, which for the purposes of this review are mentioned between brackets following the scientific name.

This review provides a brief overview of the reported role of mitochondria in the pathophysiology of NAFLD and then focuses on the therapeutic use of mushroom-enriched diets in liver lipid metabolism, oxidative stress and carcinogenic progression from a mitochondria-centric perspective.

## 2. The Molecular Pathophysiology of NAFLD

Non-alcoholic fatty liver disease (NAFLD) is thought to emerge from a state of insulin resistance (IR) as the product of an interaction between the obesogenic environment and genetically predisposing alleles (e.g., I148M in PNPLA3, E167K in TM6SF2, MBOAT7 rs641738 variant) [37,38,39]. The consumption of fat and sugar-rich, hyper caloric diets together with a lack of physical activity results in weight gain/obesity and over time can lead to a metabolic imbalance, compromising the systemic physiological processes of energy homeostasis [40]. This metabolic milieu increases the risk for the appearance of the MetS phenotype, involving dyslipidemia, hypertension and T2DM, which are strongly and bidirectionally associated with NAFLD. Whilst the presence of predisposing genetic variants seems to disassociate NAFLD from the MetS, the more lifestyle-driven NAFLD emerges secondary to obesity-induced peripheral IR, which causes an increased influx of energetic substrates into the liver [41,42,43,44]. In IR states, the signalling cascade usually elicited by binding of insulin to its receptor and the subsequent phosphorylation of IRS-1/PDK1/PI3K/Akt is impaired, compromising tissue-dependent downstream effects [45,46].

On the one hand, the white adipose tissue (WAT) becomes unresponsive to the inhibitory effect of insulin on hormone-sensitive lipase (HSL), causing postprandial lipolysis and a consequent free fatty acids (FFAs) influx into the liver [47,48]. This FFAs influx seems to be associated with an upregulation of the hepatic expression of FFA transporter CD36 [49,50]. Three-fold higher than in physiological conditions, the uptake of WAT-derived FFAs account for 60% of the intrahepatic triglyceride (IHTG) content in the steatotic liver [51,52,53]. On the other hand, IR in skeletal muscle results in a decreased GLUT4-mediated glucose disposal, burdening the liver with the need to deal with the postprandial glucose load usually metabolized by myocytes [54,55]. This hyperglycemic environment and the emerging hyperinsulinemia promote the hepatic activation of transcription factors such as carbohydrate responsive element binding protein (ChREBP) and sterol regulatory element binding protein 1c (SREBP1c), which mediate the expression of genes coding for lipogenic enzymes (e.g., acetyl-CoA carboxylase (ACC), fatty acid synthase (FAS) and stearoyl-CoA desaturase-1 (SCD-1)) [56,57]. Consequently, de-novo lipogenesis is favoured, thereby converting glucose and fructose into FFAs and accounting for 25% of the IHTG content in NAFLD [53,58]. Additionally, the decreased action of lipoprotein lipase (LPL) in IR states results in an increase in triglyceride (TAG)-enriched remnant chylomicrons reaching the hepatic cells [59]. The uptake of their cargo accounts for the remaining 15% of the lipid storage pool in NAFLD [53,59]. In this scenario, insulin-resistant WAT-derived lipolysis together with the chronic intake of high-fat, high-sugar hypercaloric diets leads to an increased influx of precursors for hepatic TAG synthesis.

The hepatocytes esterify FFAs into TAGs and store them together with cholesterol esters in lipid droplets (LDs), a mechanism that limits the lipotoxic effect of free FFAs. LDs are dynamic structures that bud out of the outer membrane of the endoplasmic reticulum (ER) once the accumulation of TAGs within the ER bilayer reaches a critical concentration [60]. IHTG content is correlated with MetS components, impaired hepatic insulin clearance and IR in WAT, but this linear relationship seems to plateau for hepatic and muscle IR after reaching 1.5% and 6% of IHTG accumulation respectively [61,62,63]. Nonetheless, steatosis reduction is still considered a primary outcome measure for NAFLD clinical trials, and interventions targeting weight loss have been shown to reduce IHTG levels, reverse hepatic IR and induce NAFLD resolution in humans [64].

Hepatic IR is characterized by a decrease in hepatic glycogen synthesis, an increase in endogenous glucose production and the paradoxical upregulation of de-novo fatty acid biosynthesis, all of which are dependent on Akt downstream effectors [45,46]. In NAFLD, lipid oxidation and secretion pathways also become upregulated in an attempt to compensate for the excessive lipid influx and the increased IHTG content [65]. LDs serve as FFA donors for very-low density lipoproteins (VLDL) lipidation in the ER and as a substrate for β-oxidation in the mitochondria [60]. The loss of insulin-dependent repression of apolipoprotein B (ApoB100) synthesis and microsomal triglyceride transfer protein (MTP) action, which are respectively responsible for the early assembling and lipidation of VLDL in the ER, results in an initial increase in VLDL secretion [66]. In parallel, FFAs act as ligands of peroxisome proliferator-activated receptor α (PPAR-α). PPAR-α, together with peroxisome proliferator-activated receptor gamma coactivator 1-alpha (PGC-1α), induces the expression of MTP and carnitine palmitoyltransferase I (CPT-1). As the rate-limiting enzyme of β-oxidation, CPT-1 overexpression increases delivery of FFAs into the mitochondria and, subsequently, hepatic β-oxidation rates [67,68,69].

Once devised as a “two-hit model”, the current “multi-hit” model posits that the pro-inflammatory environment in the liver responsible for the steatosis progression to NASH, arises from a combination of synergistic pathogenic effects occurring at different levels [70,71,72]. In the WAT, the obesity-induced chronic low-grade inflammation leads to the production and release of cytokines, such as monocyte chemoattractant protein-1 (MCP-1), interleukin-6 (IL-6) and tumor necrosis factor α (TNF-α). In the gut, diet-induced dysbiosis results in a pathological increase in gut permeability, signalling the liver with pathogen-associated molecular patterns (PAMPs), such as lipopolysaccharides (LPS). In fact, the gut–liver axis is an important factor in the development of NAFLD, including its progressive subtype non-alcoholic steatohepatitis (NASH) [73]. Human studies demonstrated that patients with NAFLD have lower gut microbiota diversity compared with healthy subjects [74]. Furthermore, cross-sectional studies with adults and childrens with NASH showed a dysregulation in the relative abundance of bacteria from the phyla Bacteroides and Firmicutes, with an increase in the Bacteroides population and a respective decrease in Firmicutes [75]. Imbalanced bacterial populations in the gut lead to an increase in bacterial metabolites that can cross the gut barrier and reach the liver, promoting inflammation and disease progression [75]. Two of these metabolites, ethanol and phenylacetic acid, are positively correlated with hepatic steatosis and NASH in human patients [76,77].

In the liver, the chronic imbalance between the lipid influx and efflux, as well as between anabolic and catabolic processes, induces mitochondrial dysfunction, possibly increased oxidative stress and the accumulation of lipotoxic FFAs. Furthermore, lipid intermediaries with signalling properties such as diacylglycerols and ceramides, resulting from partial hydrolysis of triglycerides, are thought to exacerbate IR, ER-stress and lipotoxicity-induced inflammation by cytokine formation in the liver [78,79]. Overall, WAT and liver cytokines, PAMPs, as well as damage-associated molecular patterns (DAMPs) from dying hepatocytes, initiate or perpetuate an inflammation state contributing to the detrimental impact of the abovementioned overload of energy substrates in the liver [2,80,81,82]. This proinflammatory milieu triggers the recruitment of macrophages, neutrophils and T-lymphocytes and the activation of Kupffer cells in the liver, exacerbating ROS production, oxidative stress and ER-stress, compromising organelle cross-talk and ultimately cellular function leading to inflammation, cell death, fibrosis, cirrhosis and progression to HCC [2,80,81,83].

## 3. Mitochondria Dysfunction in NAFLD

Mitochondrial dysfunction is considered one of the components of NAFLD development and progression [83,84]. Structural and functional changes as well as alterations in mitochondrial molecular composition, dynamics, as well as organelle cross-talk seem to be affected and contribute to the natural course of NAFLD (Figure 1) [65,85,86,87,88,89].

### 3.1. Metabolic Alterations: Dysfunctional TCA Cycle 

In the liver, mitochondria act as bioenergetic hubs for both anabolic and catabolic pathways such as gluconeogenesis, de-novo lipogenesis, ketogenesis and cholesterol synthesis, fatty acid β-oxidation and pyruvate oxidation, all of which converge at the tricarboxylic acid cycle (TCA) [90]. Through sequential oxidation reactions, the TCA generates reduced intermediates (NADH, succinate) that deliver electrons to Complexes I and II of the electron transport chain (ETC), where oxidative phosphorylation (OXPHOS) takes place generating an electrochemical gradient coupled with ATP production [91,92].

Upon the development of IR and NAFLD, an excess of carbon sources is delivered to hepatic mitochondria to undergo oxidation (Figure 1) [67,93]. Indeed, ^13^C-NMR-based isotopomer analysis has demonstrated that humans with NAFLD present an increased TCA cycle flux and higher mitochondrial respiratory rates and ATP turnover in the early stages of the disease as reviewed in Sunny et al. [67]. Seemingly responsible for this phenomenon is the increase in fatty acid oxidation observed in samples from NAFLD patients with simple steatosis and mediated by the increased transcription of CPT-1 [67,68,69]. This increased influx of FFAs may trigger a metabolic shift, promoting mitochondrial FFA β-oxidation over pyruvate oxidation, the so-called Randle’s cycle [84,94,95,96]. Of note, indeed NAFLD patients present a “metabolic inflexibility”, that is, a reduced capacity to switch back from FFA to glucose oxidation, even in the presence of insulin [43,46,97]. The catabolism of fatty acids through β-oxidation results in the formation of acetyl-CoA molecules that feed the TCA cycle [98].

Putatively, augmented acetyl-CoA requires a faster TCA cycle turnover to become metabolized [67,99,100]. However, high acetyl-CoA levels inhibit different enzymes such as pyruvate dehydrogenase (PDH) and branched-chain α-ketoacid dehydrogenase (BCKD). BCKD action is necessary for the conversion of branched-chain amino acids (BCAA) to succinyl-CoA, a TCA intermediate. Therefore, despite the elevated serum BCAA levels present in NAFLD patients, resulting from the loss of insulin-mediated suppression of BCAA release, their use as anaplerotic substrates is thought to be impaired, leading to “anaplerotic stress” in the liver [101,102]. Conversely, an increased acetyl-CoA pool allosterically increases the activity of pyruvate carboxylase (PC), a phenomenon observed in NAFLD, and is thought to mediate the anaplerosis of oxaloacetate. This constitutes an alternative for pyruvate to contribute to TCA flux when PDH is inhibited and is hypothesized to be the mechanism by which WAT lipolysis controls hepatic gluconeogenesis [67,103,104,105]. ^13^C-NMR-based analysis confirmed that patients with high IHTG content present a 50% increase in TCA anaplerosis and an equivalent increase in gluconeogenesis via the cataplerosis of oxaloacetate through phosphoenolpyruvate kinase (PEPCK). Futile pyruvate cycling, associated with the formation of pyruvate from either phosphoenolpyruvate or malate was also upregulated by 55% in patients with steatosis [67,105].

Concomitantly, hepatic mitochondria are also involved in other ATP-consuming anabolic pathways such as cholesterol biosynthesis or de-novo lipogenesis [67,90]. The increased levels of β-oxidation seem to result in an increase in citrate within the mitochondrial matrix that can be transported to the cytosol via the citrate-malate shuttle and converted to acetyl-CoA and oxaloacetate by the enzyme ATP-citrate lyase [90,106]. Indeed, NAFLD patients present increased citrate levels in plasma [107]. Whilst oxaloacetate can re-enter the mitochondria in the form of malate, acetyl-CoA can be used for the synthesis of ketone bodies and cholesterol, or alternatively, converted into malonyl-CoA by ACC, thereby starting FFA biosynthesis [106,108].

Human NASH mitochondria present lower mitochondrial membrane potential (∆Ψm) [12], and while the elevated TCA cycle oxidative flux persists, an increased proton leakage in respiration is observed as compared to NAFL and control patients. Furthermore, paralleling its increase across the NAFLD continuum, this observed proton leaking effect was inversely correlated with peripheral insulin sensitivity and positively correlated with IHTG content and FFAs circulating levels in plasma [69]. Physiologically, the TCA cycle is tightly coupled to mitochondrial respiration, but high nutrient availability may induce uncoupling by dissipating the proton electrochemical gradient, overall dissipating the proton-motion force that drives ATP synthesis [13,109,110]. An alternative explanation for the deficient mitochondrial respiration might be the alterations in the mitochondria lipid composition, which are already present in steatosis. Changes in the biophysical properties of the inner mitochondrial membrane have been reported in steatosis and could be associated with the functional performance of several membrane proteins. Mediated by SCD-1, the production of monounsaturated fatty acids (MUFAs) as a protective mechanism against the lipotoxic effects of palmitate produced by de-novo lipogenesis is thought to increase the membrane fluidity of mitochondria. This has been hypothesized to hinder the formation of *supercomplexes*, and thus OXPHOS efficiency. Of note, these hallmarks of mitochondrial dysfunction precede the formation of ROS and oxidative stress [65].

### 3.2. ROS Production and Oxidative Stress

The increased oxidative TCA flux in NAFLD places a redox stress burden on the mitochondrial respiratory chain, increasing the need for oxidized electron carriers [110]. The increased demand for NAD+ and FAD+, the oxidized Complex II co-factor, may interfere with the pace of migration of electrons along the ETC complexes causing electron leakage, predominantly at complexes I and III, and the production of superoxide anion radicals (O•−), resulting from univalent oxygen reduction [111]. Superoxide anion can be scavenged and dismutated into hydrogen peroxide (H_2_O_2_) by the enzyme superoxide dismutase 2 (MnSOD) present in the mitochondrial matrix. In turn, H_2_O_2_ can be converted to water by peroxiredoxins, thioredoxins and glutathione peroxidase (GPx) in mitochondria, using GSH as a co-factor. Additionally, it can diffuse across the mitochondrial membrane and be detoxified by the enzyme catalase (CAT) [112,113]. Moreover, via the Fenton reaction, H_2_O_2_ can be transformed into hydroxyl radical (HO^•^), a very reactive yet short-lived ROS species (Figure 1) [113].

As defined by mitohormesis, ROS production is physiological at low levels, acting as a crucial effector in proliferation, expression of antioxidant enzymes and insulin signalling. However, high levels of ROS formation causes oxidative stress and cell damage by reacting with its different components [90,114,115]. Oxidative stress occurs when the antioxidant capacity of the cell is not sufficient to neutralize the overproduction of ROS. ROS generation causes the peroxidation of phospholipids and cardiolipin at the mitochondrial membrane. Cardiolipin is involved in the protein folding and activity of ETC complexes, and its peroxidation leads to ETC activity impairment [116,117,118]. This can trigger a self-perpetuating cycle of ROS production causing lipid peroxidation, mitochondrial DNA damage and OXPHOS impairment leading to mitochondria dysfunction [9,10,12].

Furthermore, HO^•^ can cause the peroxidation of proteins, amino-acids, lipids and cholesterol. The peroxidation of lipids, which is particularly detrimental in a lipid-rich environment such as the steatotic liver, results in the formation of reactive aldehydic derivatives such as trans-4-hydroxy-2-nonenal (HNE) and malondialdehyde (MDA) that cause lipotoxicity. Concomitantly, the peroxidation of cholesterol produces oxysterols, a ligand for liver-X receptors (LXRs), which contributes to hepatic IR via activation of SREBP1c. ROS emergence is also associated with the activation of c-Jun N-terminal kinase (JNK) and NF-κB, further promoting IR and inflammation. Furthermore, reacting with Fe-S clusters at their active sites, ROS inactivates crucial TCA cycle enzymes and ETC Complexes, such as aconitase, succinate dehydrogenase (Complex II) and Complex I, whilst the displaced Fe moiety creates more ROS via Fenton reactions [119,120].

Moreover, ROS may induce mitochondrial DNA damage and reduce the activity of PGC-1α, the regulator of mitochondrial transcription factor A (TFAM) and nuclear factor erythroid 2-related factors 1 and 2 (Nrf-1 and Nrf-2), regulators of the expression of ETC complexes and cytoprotective mediators, including antioxidant effectors (Figure 1) [121].

All these mechanisms seem to be involved in the progression from NAFL to NASH. Indeed, NASH patients present increased ROS production, DNA damage, as measured by 8-Oxo-2’-deoxyguanosine (8OHdG) levels, and hepatic lipid peroxidation coupled with decreased expression of ETC Complexes I, III, IV and V [69,122]. Similar to obese and T2DM patients [123], both NAFL and NASH patients present decreased mRNA expression of PGC-1α, TFAM and Nrf-1, whilst only NASH patients presented increased levels of JNK phosphorylation and decreased CAT activity in the liver [69]. Indeed, the reduction of mitochondrial antioxidant defences, such as coenzyme Q10, SOD, CAT, glutathione S-transferase (GSTs) activity and GSH levels is correlated with the severity of the disease, contributing to increase mitochondrial dysfunction that may ultimately result in hepatocyte death [9,10,124]. Mitochondrial GSH (mtGSH), which represents 10–15% of the GSH pool of the cell has been described to be severely depleted in in-vivo NASH models and human patients [93,115,125,126]. The accumulation of free cholesterol in organelle membranes has been observed in NASH models and is thought to decrease the membrane fluidity of hepatic mitochondria [127,128,129]. This impairs the activity of several transporters such as SLC25A1, which mediates citrate export [130], or 2-oxoglutarate (2-OG) carrier, that exports 2-OG coupled to the import of GSH. In turn, mtGSH depletion sensitizes the hepatocytes for TNF-α-mediated cell death [93,131,132].

### 3.3. Apoptosis

As the molecular composition of mitochondria changes upon steatosis [65,93,118,131], oxidative stress increases and the oxidative capacity of the mitochondria becomes impaired, cytosolic FFAs accumulate and ATP synthesis is reduced. Furthermore, the mitochondria vulnerability to secondary hits, such as calcium, and the probability of mitochondrial transition pore (mPTP) opening increases upon steatotic insult [65,133,134]. Lipid peroxidation, mitochondrial DNA damage and OXPHOS impairment may solely or in combination lead to the aberrant functioning of ETC complexes and TCA enzymes, driving late-stage mitochondrial dysfunction [9,10,12]. FFAs further aggravate this phenotype by decreasing ATP synthesis efficacy and the formation of 4-HNE and MDA moieties and activation of the JNK-pathway, contributing to ER-stress (Figure 1) [135,136,137].

In response to the increased ROS production, mitochondrial damage and ER-stress, the cell mediates the activation of either survival or pro-apoptotic pathways, including phosphoinositide 3-kinase (PI3K)/Akt, MAPK, Nrf-2/Keap1, NF-κB, the tumour suppressor p53 and JNK [138]. JNK and p53, activated by ROS and ER-stress, induce the inhibition of B-cell lymphoma-extra-large (Bcl-xL) and the activation of pro-apoptotic Bcl-2-associated X protein (Bax/Bak). Apoptosis proceeds by mitochondrial outer membrane permeabilization (MOMP) and the opening of Bax/Bak channels, which ultimately allows for the release of the apoptosis-inducing factor (AIF), Endonuclease G (EndoG), cytochrome C (CYC) and second mitochondria-derived activator of caspase (Smac) into the cytosol [138,139]. In parallel, the negative regulation on the inhibitor of apoptosis proteins (IAPs) mediated by the translocation of a series of IAP antagonists such as Smac, HTRA2/Omi and apoptosis-related protein in the TGF-ß signalling pathway (ARTS) to the cytosol, results in the release and activation of caspases. Subsequently, caspases cleave key proteins for survival and homeostasis, such as poly (ADP-ribose) polymerase-1 (PARP-1), initiating the degradation of cellular components by proteolysis (Figure 1) [138,140,141,142]. In HCC, the presence of alterations in the expression and/or activation of p53 and the activation of pathways that ensure cell survival, such as PI3K/Akt, confer tumour cells resistance to apoptotic stimuli [140,143,144]. In cancer cells, the mitochondrial apoptosis pathway is deregulated due to elevated expression of pro-survival versus low expression of pro-apoptotic Bcl-2 family proteins. Moreover, pro-apoptotic Bax and BH3-only proteins, such as the p53-upregulated modulator of apoptosis (PUMA) and Noxa, are transcriptional targets of p53, the function of which is impaired in most cases of cancer [140,143]. HCC cells may evade apoptosis by decreasing the expression of cardiolipin [137]. ROS production and lipid peroxidation oxidize cardiolipin to form 4-HNE and other oxidized moieties. A decrease in cardiolipin and 4-HNE was observed in human samples from HCC patients [135,137]. Cardiolipin oxidation induces CYC release and thus, down regulating its expression might constitute a mechanism to avoid mitochondria-mediated apoptosis.

To sum up, mitochondria play a central role in the pathophysiology and progression of NAFLD as well as in the development of HCC, which can be a late-stage consequence of NASH. Hepatic mitochondria undergo bioenergetic remodelling to face the metabolic burden imposed by the increased FFAs load secondary to systemic IR. In turn, a decompensation of these processes may result in ROS formation and mitochondrial dysfunction, contributing to the development of NASH. Lastly, hepatic mitochondria also seem to be involved in anti-apoptotic oncogenic processes driving HCC. Targeting mitochondrial dysfunction is thus a promising approach for the treatment of the NAFLD continuum. The following section describes some of the in-vitro and in-vivo studies on the beneficial effects of mushroom-enriched diets or mushroom-derived compounds/extracts (Box 2) in preventing/reverting such liver damage.

Box 2Extraction protocols are often applied to mushroom fruiting bodies or mycelia to study their effects in cellular and animal models.The extraction, but also the mushroom species, dictate the type and amount of active compounds available. The most common extraction methods are based on aqueous (water and hot water) or alcoholic solvents [145,146]. Other organic solvents can be used, such as acetone, chloroform or ether, and different conditions applied, for instance more alkaline or acidic extractions [145,147,148]. Ethanol or methanol extracts are rich in phenolic compounds (ex. flavonoids, polyphenols, terpenoids, lignans and alkaloids), while aqueous extracts are rich in polysaccharides, proteins/peptides, lectins, glycoproteins, among others [149]. Both aqueous and organic extracts present antioxidant activities, however, organic extractions, richer in phenolic compounds, seem to have higher antioxidant capacities [150,151]. Moreover, results from cell and animal studies indicate that aqueous extracts activate immunological responses, while ethanol/methanol extracts inhibit immune cell activity and present higher cytotoxic effects in cancer cell lines [149,152,153]. Further separation and purification of fruiting bodies or mycelia extracts allows to isolate and identify specific fractions or single compounds. After aqueous or ethanol extractions, isolation of phenolic, polysaccharidic, protein and lipidic fractions can be achieved by subsequent extractions, precipitation, treatment with salt solutions, column fractionation, dialysis and/or ultrafiltration [31,154]. Purification of specific compounds can involve several steps. For example, β-D-glucans can be separated from the general polysaccharidic fraction through precipitation with two to three volumes of cold ethanol, and triterpenes through methanol extraction followed by purification in-silica gel chromatography [154,155].

## 4. Mitochondria: A Target for Steatosis Treatment 

The most effective and comprehensive therapy for the management of NAFLD is the implementation of a lifestyle intervention [156,157]. Aiming at weight loss, calorie-restricted diets and regular physical activity can improve hepatic mitochondria dysfunction by decreasing FFA liver input and alleviating oxidative stress. Indeed, physical exercise offers preventive and therapeutic effects on NASH-induced mitochondrial bioenergetic dysfunction as well as on the mitochondrial phospholipidomic profile in High-Fat Diet (HFD) animal models [158,159,160]. In turn, caloric restriction reduces oxidative stress and contributes to increasing hepatic mitochondrial biogenesis and respiratory efficiency [161,162]. Not only caloric intake but also nutrient composition is of relevance when implementing dietary regimes. The Mediterranean diet, known to decrease steatosis, is rich in natural products with bioactive properties, for instance polyunsaturated fatty acids (PUFAs) with anti-inflammatory effects [163,164,165,166].

However, the low adherence to recommended changes in dietary and exercise behavioural patterns pose the need for new alternatives [167,168]. At present, there is no pharmacotherapy approved for the treatment of NAFLD. Interestingly, drugs prescribed for the treatment of prediabetes and T2DM namely pioglitazone, metformin and liraglutide seem to provide mitochondria-mediated therapeutic effects in the context of NAFL and NASH [13,169,170,171,172,173]. Despite the positive effects on liver function, these drugs are not yet available for non-diabetic NAFLD individuals, and the search for new candidates, including natural compounds, continues [174,175].

The potential of edible mushrooms (and truffles), either as nutraceuticals or as preventive and therapeutic agents for metabolic-linked conditions, has been demonstrated in a wide range of cellular and animal models, as well as in human clinical trials [176,177,178,179,180,181]. Safety and tolerability of mushrooms extracts or isolated compounds have also been tested in animal models and human subjects, showing no adverse effects and promising results for a wide range of diseases, from not only metabolic, but also immune and viral-related conditions [182,183,184].

## 5. Mushrooms Enriched Diets Affect Liver Mitochondrial Metabolism

Mushrooms (and truffles) have low fat content and are nutritional sources of proteins, carbohydrates, low-digestible and non-digestible carbohydrates (known as dietary fibers), vitamins, minerals, and PUFAs [185]. As a substitute for red meat, mushrooms lower the energy density of diets, exerting positive effects on body weight without compromising palatability or satiety [186,187]. In fact, certain mushroom compounds/extracts seem to regulate appetite and satiation, an effect that might be related with an improvement in leptin sensitivity. Produced in a rate proportional to the amount of fat stored in the WAT, leptin decreases appetite and increases energy expenditure [188]. Obesity, T2DM and NAFLD patients are leptin-resistant and present hyperleptinemia [189]. Research on NAFLD in in-vivo models suggest that mushroom species such as *Lentinus edodes* (Berk.) Pegler *(shiitake)*, *Hericium erinaceus* (Bull.:Fr) Pers (lion’s mane / yamabushitake) and *Lepista nuda* (Bull.) Cooke (Synonym *Clitocybe nuda*; wood blewit) seem to promote at least part of their positive effects through a decrease in leptin levels. This reduction is accompanied by a lowering effect on body weight, adiposity and circulating TAGs, overall suggesting that by improving leptin sensitivity, these species might alleviate the FFAs delivery to the liver [190,191,192].

An alternative mechanism by which mushroom-enriched diets might reduce the burdening influx of FFAs into liver mitochondria is by reducing inflammation and IR in the WAT. Supplementation of an HFD with either aqueous extracts of *Antrodia cinnamomea* T.T Chang & W.N. Chou (“niu-chang-chih”) or *Ganoderma lucidum* (Curtis) P. Karst (reishi/lingzhi) for 8 weeks induced a decrease in the WAT mRNA and protein levels of pro-inflammatory cytokines including IL-6, TNF-α and MCP-1, as well as inflammatory mediators JNK and NF-κB as compared to the control. This decrease was paralleled by a reduction in total cholesterol (TC), LDL cholesterol and TAG serum levels as well as an alleviation of IR and hepatic steatosis [193,194,195]. The reduction in WAT inflammation and IR might be responsible for the decrease in IHTG content observed in these rodents, and in turn, reduce the metabolic burden on hepatic mitochondria.

Mushrooms in the diet also present modulatory actions in the gut by reducing intestinal lipid absorption and influencing the gut microbiota. Active compounds present in the fruiting body of *H. erinaceus* seem to induce an inhibition of pancreatic lipase activity and a consequential reduction in dietary lipid absorption as demonstrated by in-vitro cultures and the increase on fecal lipid content after a 5% supplementation to a standard diet for 7 weeks in a mouse model [192]. Similarly, supplementation with 5% chitosan (the deacetylated form of chitin) derived from *Agaricus bisporus* (J.E.Lange) Imbach (portobello) to HFD fed mice caused a significant reduction in hepatic steatosis, adiposity index, leptinemia and serum lipid levels, as well as an increase in ceacal lipid content after 10 weeks. The apparent reduction in lipid absorption due to the effect of mushroom-derived dietary fibres is likely contributing to the reduction of ectopic steatosis in the liver [196]. In addition, the consumption of *A. bisporus* has also demonstrated positive effects on the gut health of humans by promoting laxation and changing microbiota composition (increase in *Bacteroidetes* and decrease in *Firmicutes*) [197].

The benefits of mushrooms on gut microbiota are mainly attributed to the bio-effects of low-digestible and non-digestible carbohydrates (including chitin and α/ß-D-glucans). These are the major components of the fungal cell wall, accounting for 35% up to 70% of mushrooms dry weight [28,198]. The administration of both aqueous and ethanol extracts of *G. lucidum,* as well as an aqueous extract of *A. cinnamomea* as a supplement to an 8-week HFD regime, resulted in a significant amelioration of the HFD-induced gut dysbiosis observed in the control groups fed an HFD. Ethanol extracts, contrary to aqueous extracts, are poor in polysaccharides but rich in polyphenols, which are known to have beneficial effects against NAFLD as prebiotic molecules [199]. By modulating the *Firmicutes*/*Bacteroidetes* ratio, these extracts seemed to promote the integrity of the intestinal barrier and reduce LPS translocation/endotoxemia and the consequential PAMPs-induced inflammation. This, in turn, could contribute to the observed decrease in inflammation and IR in the WAT, as reported above. Furthermore, the results from these studies seem to indicate a close association between gut microbiota composition and lipid metabolism. The beneficial changes in gut microbiota composition upon mushroom feeding were correlated with an improvement of the serum lipid profile, namely a decrease in TC, LDL and TAGs and an increase in HDL cholesterol [193,194,195,200]. Recently, similar effects have been demonstrated for metformin, which improved “leaky gut”, inducing microbiota changes and decreasing endotoxemia in a NAFLD mouse model [201].

Mushroom extract-enriched diets also showed lipid metabolism-modulating properties in the liver [202]. These effects seem closely associated to the function of adiponectin, which enhances fatty acid β-oxidation by activating liver AMP-activated protein kinase (AMPK) and PPAR-α [203]. Furthermore, AMPK mediates an increase in PGC-1α and the inactivation of enzymes involved in lipid synthesis (ACC-1, FAS and SREBP-1c), inhibiting lipogenesis and promoting β-oxidation. Indeed, metformin is thought to exert its inhibitory effect on hepatic gluconeogenesis via a reversible inhibition of mitochondrial Complex I, resulting in the activation of AMPK and linked pathways such as glycolysis, fatty acid oxidation and mitochondrial biogenesis [204,205]. Several mushroom species seem to display similar effects. An aqueous extract from *Panellus serotinus* (Pers.) Kühner (mukitake/late oyster) induced a decrease in IHTG content and hepatic injury serum markers after 4 weeks feeding in *db/db* mice. Besides a reduction in hepatic steatosis, these animals also presented an increase in adiponectin and a decrease in MCP-1, in both serum and perirenal WAT mRNA levels, as well as a significant decrease in FAS and malic enzyme activities and an increase in mitochondrial CPT-1 activity. This may lead to a decrease in lipogenesis and a concomitant increase in β-oxidation that could explain the reduction in IHTG content [203]. Similarly, the supplementation with a 1% aqueous extract of *A. cinnamomea* for 8 weeks reduced the expression of leptin and increased the expression of adiponectin, which was accompanied by an increase of AMPK and PGC-1α and a reduced expression of ACC, FAS and SREBP-1c in WAT of HFD-fed mice [193]. Conversely, ethanol extracts of *H. erinaceus* fed to an HFD mouse model elevated mRNA expression of lipogenic genes, such as SREBP1c and ACC. These extracts also elevated mRNA levels of genes regulated by PPAR-α, namely, acyl-CoA dehydrogenase (ACAD) and fatty acid transport protein 1 and 4 (FATP-1; FATP-4), whilst promoting the increase of Apolipoprotein A1 (APOA1) and LPL levels. The authors proposed that the *H. erinaceus* ethanol extract might have a PPAR-α agonist activity, therefore promoting fatty acid β-oxidation [206]. Similarly, the beneficial effects of pioglitazone on the liver are believed to occur primarily through PPAR-mediated agonist activity that promotes adipose tissue insulin-sensitivity improvement, resulting in reduced FFA delivery to the liver [207,208]. This reduction may alleviate hepatic mitochondrial dysfunction through decreased TCA cycle fluxes [207]. In fact, pioglitazone reduced hepatic TAG content, glucose and insulin levels in plasma, improving hepatic, muscle and adipose tissue insulin-sensitivity in patients with prediabetes or T2DM and biopsy-proven NASH [175]. In mice with Streptozotocin (STZ)-induced liver damage (Box 3), *L. nuda* aqueous extract, as an adjuvant to an 8-week HFD diet, induced a reduction in IHTG content that was similar to the positive control group, treated with rosiglitazone, a PPAR-agonist antidiabetic drug. This extract induced an upregulation of PPAR-α mRNA levels and an increase in p-AMPK, paralleled by a decrease in the hepatic expression of gluconeogenic genes and an increase in GLUT-4 protein levels in liver and muscle tissue respectively [209]. *Ganoderma lucidum* ethanol extract also increased p-AMPK and p-ACC (inactive form) protein content in both cellular models (HepG2 and 3T3-L1) and HFD-fed mice after 16 weeks of feeding [210]. Whilst these animals presented a significant reduction in hepatic lipid droplets and hepatic injury markers as compared to the control group, treated HepG2 cells presented lower lipid accumulation upon an FFA challenge. Additionally, a specific fraction of *G. lucidum* aqueous extract fed to a STZ-mice model for 4 weeks induced an increase in the hepatic p-AMPK/AMPK ratio suggesting an activation of the AMPK protein. Moreover, the hepatic expression of gluconeogenic enzymes, including PEPCK, was downregulated in *G. lucidum*-treated mice [211].

In conclusion, despite a still existing lack of knowledge on the direct effects of mushroom components on mitochondrial function in NAFLD, ample evidence suggests that, through a wide range of mechanisms, their bioactive compounds alleviate the FFAs influx into the liver and promote AMPK and PPAR-mediated processes. As a result, hepatic lipogenesis and gluconeogenesis are downregulated and energy expenditure processes upregulated. These effects, in turn, result in a reduction of hepatic steatosis, which suggests a promising role for therapeutic mushroom compounds in NAFLD and MetS.

Box 3Hepatotoxic compounds used for hepatic injury models include Streptozotocin (STZ), Carbon tetrachloride (CCl_4_) and D-Galactosamine (D-GalN).STZ is a well-known diabetogenic agent used for in-vivo modelling of type 1 diabetes mellitus. This compound is a cytotoxic glucose analogue that, entering via the low-affinity glucose transporter 2 (GLUT2), accumulates in pancreatic β-cells and causes insulin secretion inhibition. Since hepatocytes also express GLUT2, STZ can cause liver damage via DNA methylation, nitric oxide production, lipid peroxidation and ROS generation in hepatocytes [212,213]. Raza et al., demonstrated that mitochondria from HepG2 cells are sensitive to STZ treatment, displaying alterations in mitochondrial membrane potential and enzyme activities, resulting in ATP synthesis inhibition. Furthermore, ROS-sensitive mitochondrial aconitase activity was markedly inhibited, suggesting increased oxidative stress in STZ-induced mitochondrial toxicity [214]. CCl_4_, on the other hand, promotes liver damage and fibrosis through several pathways. Dong et al., studied the mechanisms involved in CCl_4_ toxicity in rats using proteomics and transcriptomics analysis, reporting histopathological changes and alterations in oxidative stress, inflammatory response and extracellular matrix organization [215]. D-GalN induces liver damage in a way that resembles human viral hepatitis and is frequently used in combination with LPS to model acute liver failure. D-GalN impairs RNA synthesis and induces TNF-α-mediated cell-death. Increased production of ROS has been reported in primary cultures of rat hepatocytes induced by D-GalN, leading to oxidative stress and apoptosis or necrosis [216,217].

## 6. Mitochondria: A Target for NASH Treatment 

NASH patients with advanced fibrosis have the highest risk of progression to cirrhosis and HCC, therefore posing a most profound economic impact on NAFLD healthcare [218,219]. Consequently, the development of pharmacological agents to treat NASH, is a main focus of several ongoing clinical trials.

Improving the antioxidant capacity of the cell, vitamin E and C, glutathione (GSH), ursodeoxycholic acid (UDCA) and pentoxifylline (PTX) have shown beneficial effects against lobular inflammation as demonstrated by histological analysis [157,220,221]. Similarly, natural antioxidant compounds derived from plants such as resveratrol, curcumin, silymarin and butein, are currently being explored as alternatives for prevention and amelioration of NAFLD through attenuation of oxidative stress [222,223,224,225].

## 7. Antioxidative Effects of Mushrooms in Liver 

Mushrooms (and truffles) contain diverse compounds with proven antioxidant activity, ranging from phenolic compounds (flavonoids, lignans, oxidized polyphenols, phenolic acids, stilbenes and tannins) to carotenoids, polysaccharides, proteins and peptides (glutathione and ergothioneine), vitamins and derivatives (ascorbic acid, ergosterol and tocopherols), and minerals (zinc and selenium), among others [24,30,31,147,226]. In particular, the total phenolic content of edible mushrooms has been shown to be intimately related to their antioxidant capability and to their ability to scavenge free radicals [227,228,229]. By elevating antioxidant levels in hepatocytes, mushrooms may alleviate oxidative stress damage in NAFLD.

Several in-vitro studies have demonstrated the capacity of mushroom extracts, either from fruiting bodies or mycelia, to counteract oxidative stress by radical scavenging (OH^•^, O2^•^^−^, H_2_O_2_ and DPPH), lipid peroxidation inhibition, Fe^3+^ ions reduction and Fe^2+^ ion chelation activity [24,177,230,231,232,233]. Such chelation of Fe ions, e.g., originating from disoriented Fe-S clusters of TCA enzymes or ETC proteins, may directly lower Fenton reaction-based ROS. Therefore, these studies suggest a pivotal capacity of mushroom extracts to counteract the detrimental oxidative damage of mitochondria in NAFLD. Interestingly, a *Pleurotus eryngii* (DC.) Quél. (king oyster) zinc-enriched mycelia showed a higher reducing power, DPPH and OH^•^ radical scavenging ability in vitro, than non-enriched mycelia, which might indicate an additional positive effect in the normalization of oxidative stress due to this mineral [232]. Indeed, Zinc and Selenium supplementation have shown positive effects against NAFLD, namely an improvement in the lipid profile, reduced hepatic injury markers and an amelioration of histological parameters [234]. Such findings demonstrate the positive impact of mushroom extracts against oxidative stress, but also suggest their potential utilization as mineral or metal sources. This may even enhance their therapeutic antioxidant action, as metals serve as cofactors of antioxidant enzymes and are essential for the activity of several ETC proteins and TCA enzymes [235,236].

The antioxidant properties of mushrooms have further been demonstrated in studies using rodent models (Table 1). These in-vivo models used to test the antioxidant effect of mushrooms on NAFLD-related conditions are diverse, including dietary models, hepatic injury models (Box 3), genetic models or a combination thereof. In these studies, genetic and dietary models (e.g. HFD, high sugar diet (HSD), high cholesterol diet (HCD)) closely mimic MetS-related features of NAFLD. Whereas, other models represent apparent hepatic injury (induced by either STZ, carbon tetrachloride (CCl_4_), or D-galactosamine (D-GalN)) that may provoke liver damage by inducing diabetes or fibrosis. Importantly, a consistent finding in studies using mushroom species in these diverse experimental models is the prominent increase in the activity of antioxidant enzymes (CAT, SOD, GSH, GPx) and non-enzymatic (Vitamin C and E) antioxidant levels. Consequently, and in agreement with these findings, a decrease in lipid peroxidation, usually measured by MDA levels, is ubiquitously observed (Table 1).

Such mushroom-derived effects are comparable to those of the glucagon-like peptide-1 (GLP-1) analogue liraglutide, which was shown to increase CAT and SOD2 mRNA levels while decreasing MDA levels in an HFD-fed mice model [250]. Interestingly, boosting antioxidant pathways of the diverse defence systems rather than just upregulating a singular component, seems critical for the success of antioxidant therapies in NAFLD. Indeed, selectively enhancing the scavenging activity of SOD2 without replenishing the pool of mtGSH leads to H_2_O_2_ overproduction, which seems to exacerbate NASH. H_2_O_2_ over-production may open the mPTP, while its transmembrane diffusion to the cytoplasm may even result in highly detrimental OH^•^ formation. [93,131,132]. In contrast, the capacity of mushroom extracts from species such as *Pleurotus ostreatus* (Jacq.) P. Kumm. (oyster mushroom) or *G. lucidum* to elevate the entire antioxidant defence system of hepatocytes, seems a more promising therapeutic effect against the oxidative stress in NASH.

*Ganoderma lucidum* has been shown to reverse ETC complex damage caused by ROS-induced cardiolipin peroxidation, to prevent the pro-apoptotic release of CYC and to increase TCA enzyme activities in models of aging, wound-healing and cardiovascular failure [251,252,253,254]. Such evidence further supports the potential of *G. lucidum* extracts in reversing mitochondrial dysfunction in NAFLD. Besides the increase in antioxidant activity, some studies are starting to uncover the modulating activity of mushrooms on antioxidant gene expression. Kelch-like ECH-associated protein 1 (Keap1) facilitates the ubiquitination and subsequent proteolysis of Nrf-2, a key controller of the redox homeostatic gene regulatory network. When oxidative stress increases in the cell, Nrf-2 is released and translocates to the nucleus to promote the transcription of intracellular antioxidants. Nrf-2 is also known to directly regulate ROS homeostasis and mitochondria biogenesis by promoting nuclear respiratory factor 1 (Nrf-1) transcription. Nrf-2, Keap1, NF-kB, TNF-α and IL-6 mRNA levels of STZ- treated mice were normalized to control levels after treatment with an aqueous extract from *Tuber melanosporum* Vittad. (black truffle), demonstrating decreased oxidative stress and inflammation in the liver. This effect was paralleled by an improvement of the total antioxidant capacity (T-OAC), CAT and SOD activity and vitamin C and E content in liver, which is comparable to the level of the positive control, treated with glibenclamide, a drug prescribed for T2DM (Table 1) [243,255].

In the same in-vivo model, a purified selenium-enriched polysaccharide fraction from *Catathelasma ventricosum* (Peck) Singer (imperial cat) increased the antioxidant activity in the liver and improved the lipid profile, particularly TC, TG and HDL-C, reflecting the ability to maintain tissue integrity and reverse liver damage. Of note, these selenium-enriched mycelia promoted a stronger improvement in terms of oxidative stress than the positive control glibenclamide treated-group [240,241,242] (Table 1).

Thus, mushrooms seem to have the potential to alleviate the burden of oxidative stress induced by hepatic mitochondrial dysfunction, possibly in part through activation of Nrf-2. The consequential decrease in ROS levels, also by an increased scavenging activity of free radicals, is paralleled by a decrease in lipid peroxidation. These aspects may further lead to a reduction in liver inflammation levels, and, as whole, ameliorate NASH symptoms.

## 8. Mitochondria: A Target to Prevent HCC

As described above, environmental modifiers (diet, lifestyle and gut microbiota) and genetic susceptibility can worsen NAFLD pathology to fibrosis, decompensated cirrhosis or HCC [7]. Such severe liver disease stages are approached by intense/aggressive treatments that unfortunately, however, only offer modest effects on median survival times, and are mostly associated with a pronounced decrease in the patient’s quality of life [256,257]. Consequently, liver transplantation still remains the most effective treatment. One of the biggest obstacles in HCC treatment is the decision of when to use which type of therapeutic option. Post-transplant complications can mean a step back in disease treatment and less invasive surgical options such as resection and ablation may be performed. However, patients with advanced or cirrhotic HCC are not eligible for these therapeutic options. Also, radiation regimes can be applied to increase patient survival in a palliative setting [257].

Pharmacological options to treat HCC are very limited. The multi-kinase inhibitor sorafenib is the only drug approved and it is mostly prescribed to patients that are non-eligible for resection. New compounds, similar to sorafenib, showed promising results in clinical trials, namely lenvatinib, and the second-line proposed agents regorafenib and cabozantinib [7,257]. However, the resistance to mitochondrial-mediated cell-death and the enhanced proliferative capacity of tumour cells renders the current pharmacological options for HCC treatment less efficient [7]. Thus, new therapeutic strategies that selectively promote apoptosis in tumour cells have the potential to be an alternative/additional approach to treat HCC.

In this line of research, novel therapies aim to target apoptosis via mitochondria, using molecules that mimic BH3 proteins and disrupt the interactions of pro-apoptotic and anti-apoptotic proteins. Bcl-2/Bcl-xL inhibitors have already been used in clinical trials and showed promising results as single or combined therapy [142,258].

Nonetheless, the relevance of both established and hypothetical strategies to treat HCC remains unclear in NAFLD-associated HCC patients [7].

## 9. Pro-Apoptotic Effects of Mushrooms in HCC 

Mushroom extracts or isolated compounds have the capacity to induce apoptosis in HCC cell lines and in vivo xenograft models via the mitochondrial pathway (Table 2). Both aqueous and ethanol extracts, or isolated compounds (GL22 from *Ganoderma leucocontextum* T.H Li, W.Q. Deng, Dong M. Wang & H.P. Hu) increased the pro-apoptotic Bax to anti-apoptotic Bcl-2/Bcl-xL ratio. This facilitates the induction of MOMP and subsequent CYC, HtrA2/Omi and Smac release into the cytosol, leading to a decrease in ∆Ψm and the activation of caspases [259,260,261,262,263,264]. PARP cleavage was also observed upon treatment with either isolated compounds or extracts [259,261]. Thus, the activation of mitochondrial-related apoptosis pathway leads to cell death in the HCC cell lines and tumour size regression in in-vivo xenograft models.

Alterations in the PTEN/PI3K/Akt pathway, such as activation of oncogenes, gene amplification and inactivation of tumour suppressors, commonly occur in many human cancers, promoting growth, proliferation and survival [267]. Mushroom extracts rich in polysaccharides from *G. lucidum, Phellinus linteus* (Berk. & M.A Curt.) Teng (meshimakobu), *Auricularia auricula* (Bull.) J. Schröt (judas’s ear mushroom) and *Pleurotus pulmonarius* (Fr.) Quél. (lung oyster mushroom) fruiting bodies suppressed the PTEN/PI3K/Akt pathway in HCC cellular models through a decrease in Akt activity and an increase in PI3K and p-PTEN levels [262,266]. Targeting of other crucial cell survival mediators (PPARα, PPARγ and protein kinase C (PKC)) and up-regulation of p53 were observed in an HCC cellular model (Huh7) upon treatment with an extract rich in triterpenes and an isolated triterpene (GL22) from *Ganoderma leucocontextum* [263].

The antitumorigenic effects of mushroom extracts and isolated compounds have also been demonstrated in in-vivo xenograft models, resulting in tumour size reduction and increased animal survival rates (Table 2). Furthermore, in the HCC Huh7 xenograft mice model, fatty acid binding proteins (FABPs) were down-regulated by GL22 [263,265]. Therefore, the authors hypothesized that the antitumoural effects of this triterpene isolated from *G. leucocontextum* might be related with alterations in lipid metabolism that promoted FFAs storage in lipid droplets and consequent immobilization. Tumour cells have high demand for FFAs in order to synthesize new biological membranes [268,269]. Thus, alterations in lipid homeostasis upon GL22 treatment lead to the decrease in lipid synthesis, namely cardiolipin, with a subsequent decrease in oxygen consumption, ATP production and the loss of ∆Ψm. Consequently, the release of CYC from the inner mitochondrial membrane promotes the activation of apoptosis-signalling cascades, resulting in cell death [263].

Therefore, the mechanisms by which mushroom extracts or isolated compounds induce mitochondrial-related apoptosis pathways are diverse and may be related with specific bioactive compounds. Modulation of pathways crucial for cell survival and alterations in lipid homeostasis seem to be related with the pro-apoptotic effects observed in HCC cell lines and in in-vivo xenograft models.

## 10. Conclusions

New therapies need to be developed to target NAFLD and NASH, which are quickly becoming the leading causes of end-stage liver disease and HCC itself, as current treatments are highly unsatisfactory. Molecular and functional mitochondrial alterations are key features in these liver diseases, pinpointing these organelles as preferable targets for new pharmacological and non-pharmacological therapies.

Beneficial effects of mushroom-enriched diets and their isolated compounds against NAFLD and related comorbidities have been demonstrated in cellular and in in-vivo models. As compiled here, such interventions ameliorate oxidative stress, hepatic lipid profiles, and reduce inflammation. This is achieved by modulating gut microbiota, nutrient uptake, lipid metabolism and the antioxidant activity of the cell, but also by amelioration of mitochondrial dysfunction in liver disease. Conversely, alternative mushrooms extracts/compounds facilitate mitochondrial-mediated apoptosis in HCC tumour cells.

The therapeutic use of mushrooms offers great versatility, as it beneficially affects metabolism and reduces inflammation and oxidative stress. Whilst pharmacological candidates for NASH target these processes, or fibrogenic pathways, individually, the administration of mushrooms as a dietary supplement can offer synergetic beneficial effects. Indeed, most of the studies reviewed here report concomitant and possibly synergistic effects of mushrooms on the gut, the WAT and the liver. This distinct property of mushroom-based therapy or -containing diet is especially relevant in the multifactorial context of NAFLD and especially NASH, where systemic synergistic metabolic alterations need to be addressed. Contrasting the more holistic approaches using extracts, which also may involve some degree of variability, the isolation of bioactive compounds from either the mycelia or the fruiting body of mushroom species has the potential of selectively targeting specific molecular effectors and could become potential candidates for the development of new drugs for NASH treatment.

This growing interest in the metabolic, but also therapeutic effects of mushrooms calls for carefully designed studies to identify their respective active compounds and to unravel their specific underlying molecular effects, as well as their possible interactions. Considering the relevance of mitochondrial dysfunction in NAFLD progression, such studies may especially focus on mitochondria as a very promising area of research and possible intervention.

## Authors Contributions

A.F. and M.A.-P. wrote the article in equal contribution. H.Z. and A.M.A. contributed in article planning, writing and supervision. P.J.O. and J.R.-S. contributed with article supervision.

## Figures and Tables

**Figure 1 ijms-20-03987-f001:**
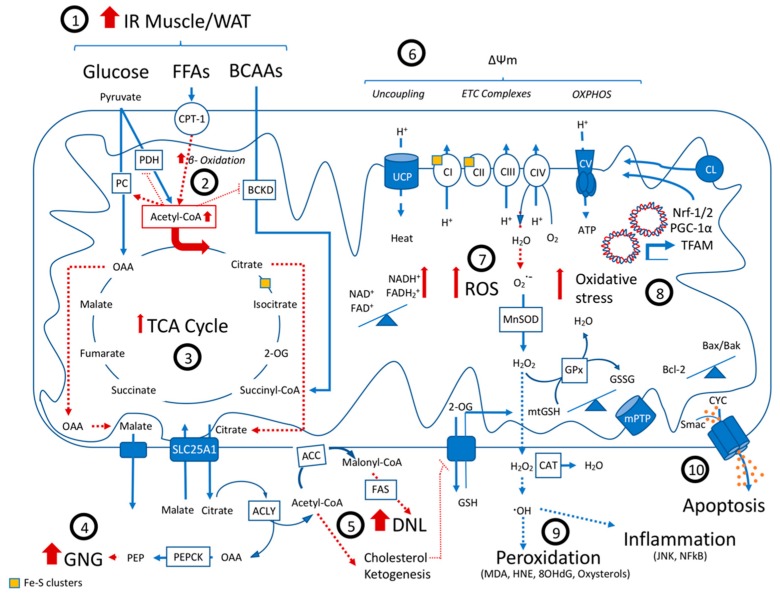
Mitochondrial dysfunction in Non Alcoholic Fatty Liver Disease: A state of peripheral insulin resistance results in an increased hepatic influx of BCAAs, glucose and especially FFAs (1), leading to an upregulation in beta-oxidation and an increase in TCA cycle flux (2). Through allosteric mechanisms, the increase in the mitochondrial Acetyl-CoA pool promotes the activity of PC but inhibits BCKD and PDH, turning the TCA cycle dysfunctional (3). The cataplerosis and export of OAA and citrate result in an increase in ATP-consuming anabolic pathways, namely gluconeogenesis (GNG) (4), de-novo lipogenesis (5), ketogenesis and the synthesis of cholesterol. Changing the lipid membrane composition and biophysical properties of the inner mitochondrial membrane, these newly synthesized lipids decrease OXPHOS efficiency and increase mitochondrial vulnerability to additional stressors (6). The increased TCA cycle flux leads to a redox stress and the need to oxidize NADH and succinate, which might increase uncoupling, but also electron leakage and the production of ROS (7). In turn, the continuous production of ROS and mtGSH depletion secondary to membrane fluidity changes might cause oxidative stress (8). As a consequence, lipid peroxidation, DNA damage and inflammation might occur. ROS also causes cardiolipin peroxidation and suppression of antioxidant gene expression, affecting ETC function (9). After a point of no return, the cumulative detrimental effects of ROS induce mitochondria-mediated apoptosis (10). Abbreviations: IR, insulin resistance; WAT, white adipose tissue; FFAs, free fatty acids; BCAAs, branched-chain amino acids; CPT-1, carnitine palmitoyltransferase-1; PDH, pyruvate dehydrogenase; PC, pyruvate carboxylase; BCKD, branched-chain α-ketoacid dehydrogenase; OAA, oxaloacetate; 2-OG, 2-Oxoglutarate; ACC, Acetyl-CoA carboxylase; TCA, tricarboxylic acid cycle; ACLY, ATP-citrate lyase; GNG, gluconeogenesis; PEP, phosphoenolpyruvate; PEPCK, phosphoenolpyruvate carboxykinase; FAS, fatty acid synthase; DNL, *de novo* lipogenesis; GSH, glutathione; UCP, uncoupling protein; ETC, electron transport chain; CI-V, complex I to V of the ETC; OXPHOS, oxidative phosphorylation; ATP, adenosine triphosphate; NAD+ and NADH2, oxidized and reduced forms of nicotinamide adenine dinucleotide; FAD+ and FADH2, oxidized and reduced forms of flavin adenine dinucleotide; ROS, reactive oxygen species; O•−, superoxide anion radicals; H_2_O_2_, hydrogen peroxide; H_2_O, water; O_2_, oxygen; HO^•^, hydroxyl radical; MnSOD, superoxide dismutase enzyme 2; GPx, glutathione peroxidase; GSH, glutathione; mtGSH, mitochondrial GSH; GSSG, glutathione disulfide; mPTP, mitochondrial permeability transition pore; CAT, catalase; MDA, malondialdehyde; HNE, trans-4-hydroxy-2-nonenal; 8-OHdG, 8-Oxo-2’-deoxyguanosine; JNK, c-Jun N-terminal kinase; NF-κB, nuclear factor kappa B; CYC, cytochrome C; IAPs, inhibitor of apoptosis proteins antagonists; Bcl-2, B-cell lymphoma 2; Bax, Bcl-2-associated X protein; Bak, BCL2-antagonist/killer 1; TFAM, mitochondrial transcription factor A; PGC-1α, peroxisome proliferator-activated receptor gamma coactivator 1-alpha; Nrf-1 and Nrf-2, Nuclear factor erythroid 2-related factors 1 and 2; CL, cardiolipin.

**Table 1 ijms-20-03987-t001:** Antioxidant effects of mushroom-enriched diets in the liver of rodents.

Species	Extract/ Compounds	Animal Model	Model	Trial Duration	Dose	Oxidative Stress Markers	Reference
*Pleurotus ostreatus* (FB)	Polysaccharides	Wistar male rats (7 weeks of age)	STZ-induced DM + HFD	4 weeks	100–400 mg/kg	↑ CAT, ↑ SOD, ↑ GPx, ↓ MDA*	[237]
*Pleurotus ostreatus* (FB)	Ethanol extract	Wistar male rats	CCl_4_-induced hepatic injury	5 days	200 mg/kg	↑ CAT, ↑ SOD, ↑GSH,↓ MDA	[238]
*Ganoderma lucidum* (FB)	Polysaccharides	C57BL/6 male mice at (10–12 weeks of age)	STZ-induced DM	4 weeks	60–180 mg/kg	↑ CAT, ↑ SOD, ↑ GPx, ↑ GSH, ↑ Vitamin C and E, ↓ MDA	[239]
*Catathelasma ventricosum* (M)	Se-enriched polysaccharides	ICR *** male mice	STZ-induced DM	5 weeks	100 mg/kg	↑ CAT, ↑ SOD, ↑ GPx, ↓ MDA	[240]
*Catathelasma ventricosum* (M)	Polysaccharides		STZ-induced DM	5 weeks	500/2000 mg/kg	↑ CAT, ↑ SOD, ↑ GPx, ↓ MDA	
*Catathelasma ventricosum* (M)	Se-enriched	ICR *** male mice	STZ-induced DM	5 weeks	400 mg/kg	↑ CAT, ↑ SOD, ↑ GPx, ↓ MDA	[241]
*Catathelasma ventricosum* (M)	Glucopyranose-rich heteropolysaccharides	ICR *** male mice	STZ-induced DM	5 weeks	100 mg/kg	↑ CAT, ↑ SOD, ↑ GPx, ↓ MDA	[242]
*Ganoderma lucidum* (FB)	Polysaccharides	Sprague-Dawley male rats (8 weeks of age)	STZ-induced DM	8 weeks	200 mg/kg	↑ CAT, ↑ SOD, ↑ GPx **	[230]
*Tuber melanosporum* (FB)	Aqueous extract	Wistar male rats	STZ-induced DM	6 weeks	400/600 mg/kg	↑ CAT, ↑ SOD, ↑ Vitamin C and E	[243]
*Pleurotus eryngii* (FB)	Polysaccharides	Mice	CCl_4_-induced hepatic injury	4 weeks	100–400 mg/kg	↑ SOD, ↓ MDA	[244]
*Lactarius deterrimus* (FB)	Ethanol extract	Wistar rats (8 weeks of age)	STZ-induced DM	4 weeks	60 mg/kg	↑ CAT, ↑ GSH, ↑ SOD *	[245,246]
*Grifola frondosa* (FB)	α-glucans	C57BL/6J and KK-Ay mice	KK-Ay mice	2 weeks	150/450 mg/kg (twice a week)	↑ SOD, ↑ GPx (n.s at 150 mg/kg), ↓ MDA	[247]
*Pleurotus eryngii* (FB)	Polysaccharides	Kunming male mice	HF	6 weeks	200–800 mg/kg	↑ SOD, ↑ GPx, ↓ MDA	[248]
*Grifola frondosa* (FB)	n-hexane extract	C57BL/6J mal mice	STZ-induced DM + HFD	2 weeks HFD pretreatment + 1 week treatment	300/600 mg/kg	↓ GPx, ↑ MDA *	[249]
*Ganoderma lucidum* (FB)	Peptides	Kunming male and female mice	(D-GalN)-induced hepatic injury	Pretreatment 2 weeks	60–180 mg/kg	↑ SOD (n.s at 60 mg/kg), ↓ MDA	[216]

FB: Fruiting bodies; M: Mycelia; HF: High fructose; Se: Selenium; n.s: non-significant; * blood values; ** pancreas values, *** Institute of Cancer Research; ↑ and ↓refer respectively to a relative increase or decrease in protein levels in the case of catalase (CAT), superoxide dismutase (SOD) or glutathione peroxidase (GPx)) or concentration (for glutathione (GSH) and malonaldehyde (MDA) as compared to the control group.

**Table 2 ijms-20-03987-t002:** Pro-apoptotic effects of mushroom compounds/extracts in liver cancer via mitochondrial pathway in HCC cellular and rodent xenograft models.

Species	Extract/ Compound	*In vitro*/ Animal Model	Trial Duration	Dose	Results	References
*Agaricus blazei* (FB)	Blazeispirol A	Hep 3B	3–48 hours	1–5 µg/mL	Casp9 and 3 activations, PARP degradation, ↓ Bcl-2 and Bcl-xL expressions, ↑ Bax expression, ↓ ∆Ψm. HtrA2/Omi and AIF release.	[261]
*Ganoderma leucocontextum* (FB)	Triterpene (GL22)	Huh7.5	3–24 hours	7.5–40 µM	↓ ATP-aerobic linked production, ↓LP and ↓ cardiolipin. CYC release, ↑ Bax/Bcl-2 ratio and up-regulation of p53. ↓ expression of FABPs. Casp3, 8, 9 and PARP cleavage. ↓ FABP4, PPARα, and PPARγ mRNA	[263]
		Huh7.5 xenograft (BALB/C nude male mice (4 weeks of age))	1 week	50 mg/kg	↓ Tumour size. ↓PPARα, PPARγ, FABP1, 4, and 5 expression	
*Ganoderma lucidum* (FB)	Triterpene-rich extract	Huh-7, Chang liver cells *	4–48 hours	50–200 µg/mL	↓ PKC activity. Activation of JNK and p38 MAP kinases.	[265]
*Ganoderma lucidum, Phellinus linteus, Auricularia auricula* (FB)	Polysaccharides-rich extract	HepG2, Bel-7404	24–72 hours	0.25–2 mg/ml	↓ AKT activity, ↑ PI3K and p-PTEN. ↓ Bcl-2 family protein levels. CYC and Smac release. Casp3 and 9 cleavage.	[265]
*Grifola frondosa* (FB)	Polysaccharides-rich extract	HepG2, HL-7702 *	24 hours	100–500 µg/mL	↓ Bcl-2 and ↑ Bax expression/mRNA levels. ↓ ∆Ψm, ↑ CYC, casp3 and 9 protein level.	[264]
*Pleurotus ferulae* (FB)	Ethanol extract	HepG2, H22	24–72 hours	1.368–8.208 µg/mL (flavonoids)	↑ p-JNK. In H22 cells: ↑ ROS and ↓ levels of MMP-2 and -9. ↓ ∆Ψm, ↑ Bax/Bcl-2 ratio. CYC release, cleavage of casp3, 7, 9, 12 and PARP	[259]
		H22 xenograft	8 weeks	2.74 or 5.48 mg/kg (flavonoids)	↓ Tumour size. ↑ survival rate.	
*Pleurotus nebrodensis* (FB)	Polysaccharides-rich extract	HepG2	48 hours	12.5 –125 µg/mL	↓ ∆Ψm, ↑ Bax/Bcl-2 ratio, CYC release. Casp3 and 9 activations	[25]
		HepG2 xenograft (Kunming male mice (6–8 weeks of age))	4 weeks	12.5 –125 mg/kg bw	↓ Tumour size. ↑ Bax/Bcl-2 ratio, CYC release	
*Pleurotus pulmonarius* (FB)	Polysaccharides/protein-rich extract	Huh7, Hep3B, WRL-68 *	24 and 48 hours	25–400 µg/mL	casp3 and PARP cleavage. Suppression of PI3K/AKT signalling pathway and over expression of the constitutively active form of AKT (Myr-AKT).	[266]
		Huh7 xenograft (BALB/C nude male mice (6-8 weeks of age))	4 weeks	200 mg/kg (oral), 50 mg/kg (i.p)	↓ Tumour size. ↓ expression of p-AKT, p-GSK3b, Bcl-xL, ↑ expression of cleaved casp3	
*Tricholoma matsutake* (FB)	Aqueous extract	HepG2, SMMC-7721	6–48 hours	1–5 mg/mL	↑ casp3, 8, and 9 activities. ↑ ROS and ↓ ∆Ψm. ↑ cleaved-PARP and Bad levels, ↑ Bax/Bcl-2 ratio.	[260]
		HepG2 or SMMC-7721 xenografts ((BALB/C nude male mice (6 weeks of age))	14 days	1 g/kg	↓ Tumour size. ↑ cleaved-PARP, Bax and Bad expression.	

FB: Fruiting bodies; * No effects were observed in the non-cancer cell lines; ↑ and ↓refer respectively to a relative increase or decrease as compared to the control group. Abbreviations: Caspase 9 (Casp9), PARP (poly (ADP-ribose) polymerase), Bcl-xL (B-cell lymphoma-extra large), Bax (Bcl-2-associated X protein), ∆Ψm (mitochondrial membrane potential), HtrA2/Omi (high temperature requirement protein A2/ stress-regulated endoprotease), AIF (apoptosis-inducing factor), ATP (adenine triphosphate), LP (lipoprotein), CYC (cytochrome C), FABPs (fatty acid binding proteins) PPARα (peroxisome proliferator-activated receptor α), PPARγ (peroxisome proliferator-activated receptor γ), PKC (protein kinase C), JNK (Jun N-terminal kinase), MAP (mitogen-activated protein), AKT (protein kinase B), PI3K (phosphoinositide 3-kinase), p-PTEN (phosphorylated-phosphatidylinositol-3,4,5-trisphosphate 3-phosphatase), Smac (second mitochondria-derived activator of caspase), ROS (Reactive Oxygen Species), MMP-2 (matrix metalloproteinase-2), p-GSK3b (phosphorylated-glycogen synthase kinase 3 beta), Bcl-2 (B-cell lymphoma 2).

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
