# Peer review of "Antioxidant Versus Pro-Apoptotic Effects of Mushroom-Enriched Diets on Mitochondria in Liver Disease"

_ijms, 2019, doi:10.3390/ijms20163987_

Round 1
Reviewer 1 Report
The manuscript is very well written. This emerging topic is clearly explained and exhaustive.
I suggest to write a note/box/table/ something like that, in which it is explained the nomenclature of the mushrooms, for not experts. For example, Page 10, lines 11-12: Agaricus bisporus (J.E.Lange) Imbach (portobello). What does "(J.E.Lange) Imbach (portobello)" mean? The same for the other species.
Some typing errors:
Page 5, line 41-44, Page 6, lines41-49, Page 35, lines 22 and 28: the text has to be reformatted.
Page 6, line 44, mitochondrial transition pore is written as MPTP (also at page 35, line 3), while at page 8, line27, the acronym is mPTP. Please uniform.
Page 8, line 25, H2O2, H2o and O2: the number are not in subscript as elsewhere in the manuscript. Please uniform.
Page 8, line 20: shift ACC before Acetyl-CoA carboxylase.
Author Response
Response to Reviewer 1:
Thank you for your review of our paper. Our answers follow below:
Point 1: I suggest to write a note/box/table/something like that, in which it is explained the nomenclature of the mushrooms, for not experts. For example, Page 10, lines 11-12: Agaricus bisporus (J.E.Lange) Imbach (portobello). What does "(J.E.Lange) Imbach (portobello)" mean? The same for the other species.
Answer: Box 1 has been modified accordingly to incorporate an explanation on the nomenclature of mushrooms and extracted compounds. We believe this information is sufficient for non-experts to interpret and critically appraise the findings of the studies reviewed in this manuscript. We have also included a reference to the International Code of Nomenclature for Fungi.
Point 2: Grammar errors, formatting and abbreviations.
Answer: We have corrected the formatting issues pointed out in the manuscript as well as the abbreviations and key terms throughout the main text, tables and figures.
Reviewer 2 Report
The objective of this work was to review the diverse effects of mushroom enriched diets in liver disease, emphasizing those effects dependent on mitochondria.- The letter format should be reviewed.
- The bibliography section should be updated and include work from 2019.
- I suggest to review the next work:
Intestinal Microbiota Modulation in Obesity-Related Non-alcoholic Fatty Liver Disease.
- Finally, text must be revised for mistakes.
For example:
* The word Bacteroides and Formicutes sometimes had not been wroten with cursive letter.
* cytochrome c. "C" should be wroten with capital letter.
Author Response
Response to Reviewer 2:
We are grateful for your comments and suggestions. Our answers follow below:
Point 1 and 4: Format revisions and nomenclature mistakes.
Answer: The font and other formatting issues have been carefully reviewed and corrected, cursive lettering has been adopted when appropriate and the abbreviations have been modified.
Point 2: The bibliography section should be updated and include work from 2019.
Answer: We have revisited our bibliography and besides 5 citations from 2019, our references include 33 original papers and reviews from 2018. Additionally, we believe that our bibliography is considerably extensive and representative of the areas of research under revision in this manuscript.
Point 3: I suggest to review the paper “Intestinal Microbiota Modulation in Obesity-Related Non-Alcoholic Fatty Liver Disease”.
Answer: We have read the suggested paper on the different therapeutic approaches used in clinical trials to address NAFLD through the modulation of the microbiota. We have cited this recent review and referenced the use of polyphenols as prebiotics targeting the gut-liver-axis pathophysiological effects driving NAFLD, since mushrooms (especially ethanol extracts) are rich in polyphenols.